# Metrological Aspects of Assessing Surface Topography and Machining Accuracy in Diagnostics of Grinding Processes

**DOI:** 10.3390/ma16062195

**Published:** 2023-03-09

**Authors:** Wojciech Kacalak, Dariusz Lipiński, Filip Szafraniec, Michał Wieczorowski, Paweł Twardowski

**Affiliations:** 1Faculty of Mechanical Engineering, Koszalin University of Technology, Racławicka 15, 75-620 Koszalin, Poland; 2Faculty of Mechanical Engineering, Institute of Applied Mechanics, Poznan University of Technology, 3 Piotrowo St., 60-965 Poznań, Poland

**Keywords:** grinding, simulation, modeling, diagnostics, monitoring, classification, surface topography, machining accuracy, roughness

## Abstract

The paper presents probabilistic aspects of diagnostics of grinding processes with consideration of metrological aspects of evaluation of topography of machined surfaces and selected problems of assessment of machining accuracy. The processes of creating the geometric structure of the ground surface are described. It was pointed out that the distribution of features important for process diagnostics depends on the mechanism of cumulative effects of random disturbances. Usually, there is a multiplicative mechanism or an additive mechanism of the component vectors of relative displacements of the tool and workpiece. The paper describes a method for determining the classification ability of specific parameters used to evaluate stereometric features of ground surfaces. It is shown that the ability to differentiate the geometric structure of a certain set of surfaces using a selected parameter depends on the geometric mean of the differences in normalized and sorted, consecutive values of this parameter. A methodology is presented for evaluating the ability of various parameters to distinguish different geometric structures of surfaces. Further, on the basis of analyses of a number of grinding processes, a methodology was formulated for proceeding leading to a comprehensive evaluation of machining accuracy and forecasting its results. It was taken into account that in forecasting the accuracy of grinding, it is necessary to determine the deviations, arising under the conditions of multiplicative interaction of the effects of various causes of inaccuracy. Examples are given of processes in which, due to the deformation of the technological system, dependent on the position of the zone and machining force, varying temperature fields and tool wear, the distributions of dimensional deviations are not the realization of stationary processes. It was emphasized that on the basis of the characteristics of the dispersion of the deviation value in the sum set of elements, it is not possible to infer its causes. Only the determination of the “instantaneous” values of the deviation dispersion parameters allows a more complete diagnosis of the process.

## 1. Introduction

Abrasive machining processes are widely used in the manufacture of precision machine and equipment components, in the mechanical engineering, automotive, aerospace, military, micro-engineering industries, as well as in construction, mining, medicine and many other fields.

The materials machined are a very diverse set: they include steels and metal alloys, such as light metal alloys, ceramics, sintered carbides, composite materials, minerals such as granite and basalt, among others, plus glass, concrete, wood, plastics, as well as crystals of precious stones such as diamond, ruby, sapphire, emerald and many other materials with high hardness and wear resistance. 

In line with the universality of applications and the variety of materials to be machined, tools of widely varying design and structure are produced using a variety of abrasive materials and a variety of bonding agents. Abrasives are also used in abrasive blasting and in abrasive–erosive and hybrid machining. Commonly used are tools containing diamond and regular boron nitride grains, in different varieties and varied micro-cutting mechanisms [1,2], different systems for limiting temperature rise [3], and different surface topography characteristics [4,5]. Tools made of different varieties of mono- and polycrystalline electrocorundum, sintered electrocorundum, and silicon carbide are also frequently used.

The sizes of abrasive grains most often range from 0.5 µm (abrasive films, lapping and polishing pastes) to 300 µm (grinding wheels for efficient fine grinding). The volume of 1 mm^3^ of bonded abrasive tools holds from 33 to 106 abrasive grains. 

The specific energy in abrasive and erosive machining processes typically ranges from 10 to 1000 J/mm^3^ [6]. The temperature exceeds 1200 °C in micro volumes; temperature gradients in the machining zone reach up to 106 °C/s and 103 °C/mm.

The consequences of high energy intensity of manufacturing processes are significant forces and thermal phenomena, causing deformation of the workpiece, tool and machining system. Manufacturing inaccuracy is influenced not only by process energy intensity and machining power but also the local concentration of energy [7,8], the shape and location of the machining zone [9,10]. The location and shape of the contact zone of the grinding wheel surface with the workpiece surface can be determined directly by modeling the grinding process [11,12], evaluating energy [13] and grinding forces [14,15], analysing temperature fields [16] and microcutting mechanisms [17]. Intermediate assessments of the surface condition of the tool are also useful [4,18].

Phenomena occurring in the grinding zone are described by features with a short time of occurrence (approximately a few milliseconds), cover areas with small areas of local interactions (several µm^2^ in size) and occur at great frequency (0.3–10 MHz), which makes them difficult to observe experimentally.

Modeling and simulation of abrasive machining processes, on the other hand, is a complex task due to the need to generate real-time geometric models of abrasive grains in the amount of 10^6^ to 10^10^ per second of the real process. The complexity of the phenomena accompanying the processes of interaction of abrasive grains with the workpiece material and the processes of tool wear make it necessary to use many different methods of analyzing the phenomena and analyze many characteristics of the process and its results in creating the basis for modeling. 

In solving many tasks using models of a specific abrasive machining process, an important issue is the search for a model that is most consistent with the mechanism of variation of the analyzed input and output quantities. This is of great importance for process diagnostics and performance forecasting [19,20]. It is worth taking into account the stochastic nature of the processes [21,22] and the diversity of the features of the machining process in different kinematic systems [23,24,25].

In micro-cutting processes, especially in high-precision machining, a number of phenomena and factors become important in determining the results of the process. These include random size and shape of abrasive grains [26,27], random distribution of grains on the tool surface [28], complex wear processes of abrasive tips and grains [29,30], discontinuity of the process of micrograin formation (at micro- and sub-microscale), local thermal and mechanical deformation of tools and workpiece material, including linear and angular displacement of abrasive grains under the influence of cutting resistance. This is compounded by the randomness of the micro-cutting process itself. The higher the randomness frequency, the smaller the average cross-sections of the layers cut by individual cutting edges.

In abrasive micromachining, as well as in various varieties of precision grinding, the recess of the blade into the workpiece material is much smaller than the radii of the roundness of its corners and is comparable to the height of the surface roughness in the machining zone.

The variability of the recess of the abrasive grains in the workpiece material is an unfavorable but unavoidable feature of micro-cutting processes. Therefore, the specific energy of machining depends not only on the average values of the parameters of the layers cut by individual blades but also the range of variation and distribution of the values of these parameters [6,9,31].

Abrasive grains in the contact zone with the workpiece move approximately tangentially to the machined surface, and their embedment in the material is variable along the cutting path. The variability of the cavity is the result of many factors. Among the most important are:variability of the nominal cavity, which depends on the kinematic characteristics of the method,irregularities of the workpiece surface in the machining zone,local susceptibility of the workpiece material and abrasive grains [26,32],vibration of the tool and abrasive grains,significant local variation (in the grain interaction zone) of temperature rise [7,33], especially when machining materials with low thermal conductivity [2,4] at very high speeds, variation in the properties of the machined material in micro-volumes compared with the volumes of the cut layers [1,7], variability of grain load [12], phenomena in high speed grinding [17], local forms of grain wear [29,33],macro- and microcontinuity of the chip and pile-up formation process [34,35,36].

The analysis of the state of knowledge shows that the high complexity of phenomena in grinding processes requires diagnostics containing a numerous, sequential, complementary set of research tasks.

Methodologies for complex diagnostics of grinding processes usually include many issues concerning 

selection of parameters for evaluation of geometric features of the machined surface,probabilistic analysis of inaccuracy of shaping of workpieces,analysis of local stress values in the surface layer,analysis of wear processes of abrasive tools,evaluation of the condition of the active surface of tools and the selection of parameters with a high ability to differentiate grades,use of dedicated parameters for evaluating the surface properties of the grinding wheel,determination of grinding forces and specific energy,prediction of abrasive tool life,forming the basis for process optimization

## 2. Methods and Materials

### 2.1. Comments on the Inadequacy of Process Information Extraction and the Effects of Limited Data in Diagnostic Procedures

The measurement uncertainty of a particular system depends on its technical design, technical execution and proper use. This uncertainty can be reduced by lowering the sensitivity threshold, reducing systematic and random uncertainty, increasing temporal stability and independence from observer skill [37].

Diagnosis of grinding processes can be carried out using various methodologies. The first uses experimental data resulting from the acquisition of selected averaged process features and the use of operator knowledge as a basis. The second methodology additionally takes into account modeling and analysis of features that are not experimentally determinable or their acquisition would be too costly. In this second methodology with the complementary use of models of a specific process, due to the high degree of randomness, an important issue is the search for a model that is most consistent with the mechanism of variation of the analyzed input and output quantities. This is of great importance for process diagnostics and forecasting of process performance.

It is advantageous to predetermine a narrow area of changes in input quantities, close to the predicted extremum of the objective function. However, it is not reasonable to search for the best model due to the accuracy of approximation according to the adopted criterion. It may happen that in subsequent realizations of the process, for a certain adopted set of model forms, the highest accuracy is obtained for an increasingly different model form. 

This means that the definition of the model according to the characteristics of the process does not lead to the selection of a model describing the process, but only describing a set of data, which in other realizations will probably be different. Under conditions of factor independence, in the case of an additive cumulative mechanism, it can be assumed that the distribution of the sum of effects is close to normal. In grinding processes, however, the additive mechanism of cumulative effects of disturbances does not occur very often, while the tendency to use such a model is excessive.

Comments on the use of experimental models in the form of polynomial dependencies presented below, unfortunately, apply to many works and publications by various authors. The creation of a model can be of informational or cognitive interest. The search for the form of the model with the highest correlation coefficient is, of course, not always justified. 

In the case where the form of the model does not result from the characteristics of the relationship described, one can also search for a form of the model that is not sensitive to statistically valid changes in the data, for example, for the next implementation of the process. However, it can be easily shown that a certain change, even of one value in the data set (statistically justified), in polynomial dependencies can cause significant and sometimes even multiple changes in the values of the model coefficients, and even changes in the sign of the values of these coefficients.

Another mistake is determining models of additive form. Such models for values of the independent variables (some or all) equal to zero show values that often do not make physical sense. Multiplicative models do not have this disadvantage. However, it is worth noting that the values of many characteristics of machining processes and their results, with a change in process parameters or a change in tool condition, change only to a certain extent, and these changes are often asymptotically limited, from which, for example, arises the usefulness of exponential functions with a decreasing value of the first derivative. 

An effective methodology for diagnosing grinding processes may require defining a single decision criterion using the following operations:first, normalize potentially useful diagnostic criteria, advantageously using fuzzy inference and defining linguistic category membership functions denoting the favorable value of the criterion;then, evaluate the sensitivity of each criterion for the adopted decision area;select those that have the greatest sensitivity to changes in process parameters and are not highly correlated;create from them a synthetic criterion favorably as a geometric mean of the component criteria.

### 2.2. The Importance of Simulation Procedures in the Analysis of Grinding Process Features

Figure 1 shows a diagram of the structure of the grinding process simulation application developed in the MATLAB^®^ computing environment.

The developed system for modeling and simulation of grinding processes allows

selection of parameters characterizing the process (including parameters defining the features of the process of creating pile-ups and parameters describing the vibrations of the machining system),selection of parameters characterizing the tool (including parameters defining geometric features of abrasive grains, parameters describing the process of conditioning the grinding wheel, parameters characterizing the intensity of the process of wear of abrasive grains), many parameters characterizing the size and features of the surface of the workpiece,selection of parameters characterizing the range of derivation of results, including images and films of shaping the topography of the machined surface, the activity of abrasive grains in different zones of the ground surface, machining forces and power, changes in the topography of the grinding wheel surface,selection of a large number of parameters characterizing the features of the simulation process, such as features describing the random characteristics of the shape and dimensions of abrasive grains, their distribution on the surface of the grinding wheel, process parameters, features of computational models, including the formation of efflorescence and individual layers machined with active cutting edges,selection of parameters characterizing the ways of visualization, presentation of results and animation of the process,determination of attributes for the numerous sets of parameters characterizing the range of recorded data and results,generating the surface of abrasive grains and carrying out verification of their characteristics,generating the active surface of the grinding wheel and analyzing the stereometric features of the distribution of vertices,selection of kinematics of the grinding process from among different technological varieties (e.g., grinding with the circumference of the wheel, smoothing with abrasive films, grinding with the face of the wheel with a hyperboloidal active surface),determination of local (also in micro-zones) and momentary values of parameters characterizing the shaping of the surface of the workpiece (local—in different places of the grinding zone, momentary—in successive moments of the process, in fixed time intervals),determination of changes in the stereometry of the machined surface and topography of the grinding wheel surface for sets of process parameters and conditions beyond the current or standard applications,determination of local and momentary as well as global parameters characterizing the load on individual grains, the work conducted (and its local changes and changes over time), the distribution of energy fluxes,determination of the influence of tool characteristics and machining parameters and conditions (including isolated changes) on the values of local and momentary values of parameters characterizing the shaping of the surface of the workpiece,determination of the influence of process disturbances on the implementation and results of the grinding process,analysis of processes with new types of tools with zone-variable structure, with aggregate and hybrid grains, with zone- and directionally variable properties, also tools with variable susceptibility,determination of numerous data sets for analysis of stereometric features, evaluation of the suitability of new evaluation and classification parameters, and development of assumptions for favorable tool modifications and selection of machining parameters and conditions.

## 3. Results of Simulation Tests

Simulation studies of the grinding process were carried out according to the scheme depicted on Figure 1 using calculation procedures developed in the Matlab^®^ environment. The input parameters and ranges of results for the analyses included, among others, the following:grinding wheels with the following geometric dimensions: D = 250 mm, H = 10–50 mm and the type of abrasive grains made of Al_2_O_3_, regular boron nitride and diamond with grain size from 46 to 240, generated and verified by comparing vertex angles, radii of vertex rounding, flatness of the surfaces of the abrasive grains, for the assumed average distances between grains in the range of 1.2 to 2.5 grain dimensions;workpiece characterized by the assessment of the ratio of pile-ups formation during microcutting with grains of various shapes and spatial orientation and by the values of the coefficient in the formulas for microcutting forces;process parameters: longitudinal feed speed in the process of grinding flat surfaces v_w_ = 0.01–1 m/s, grinding speed v_s_ = 20–60 m/s, depth of cut a_e_ = 1–200 µm, cross feed of the table fa = 0.5–5 mm/stroke;parameters of calculation procedures: calculation resolution in the range of 0.1–0.5 µm, size of the geometry data matrix of individual models of abrasive grains, e.g., 120 × 120 cells, grinding wheel surface data matrix size $120\pi D\sqrt{n_{g}}\times 120\pi H\sqrt{n_{g}}$ cells (where $n_{g}$ is the number of grains), machined surface data matrix, e.g., 50,000 × 100,000 µm;calculation procedures for pile-ups developed from the 3D FEM analysis of single-grain micro-machining modeling processes [18];analyses of the activity of the grains in individual fragments of the grinding zone as well as tangential and normal forces acting on individual grains, taking into account the shape of the grain and the cross-section of the machined layer and the characteristics of the material;analysis of the wear of the grains, taking into account their strength, resistance to chipping and random load, which were described in work [38].

Example images of surface topography shaping in the process of grinding planes with traverse feed are shown in Figure 2 and Figure 3, where the position of the grinding wheel is marked, and graphs of grinding forces during the working pass and return are included for two selected positions of the tool. Figure 3 shows a close-up of the surface being shaped during grinding. More example results of shaping machined surfaces are posted on YouTube in the @PrecisionMechanics channel at https://www.youtube.com/@PrecisionMechanics/videos (accessed on 29 January 2013).

The developed simulation procedures greatly facilitate the determination of data sets for activity and load analysis of abrasive grains. Figure 4 shows the location of the active grains on the active surface of the grinding wheel in Zone 1 with a width equal to the value of the traverse feed. The left side of the figure shows the values of force per grain. On the right side, color maps of grain cavities in micrometers are shown. The upper part of the figure contains data on the cross-sections of the layers cut by grains and highlights the zones on the surface of the grinding wheel. These data were determined for the following grinding parameters and conditions.

The development of the test results (Figure 5, Figure 6 and Figure 7) indicates the reasons for the variation of conditions and process features in different parts of the grinding zone. This is important for interpreting the relationship between the results of diagnostic measurements and the variation of process features in different parts of the machining zone.

## 4. Bases of Parameter Selection for Evaluating Geometric Features of Machined Surfaces

Among other things, the geometric structure of a surface significantly influences the friction and wear processes of mated surfaces, rolling and sliding one on another. It influences contact deformation and stiffness, stress concentration and fatigue strength, resistance to corrosive influences and vibration damping. 

The surface topography also determines the tightness of connections, contact resistance, contact heat conduction, magnetic properties, the phenomena of reflection, absorption and transmission of waves (light, electromagnetic, etc.) [39].

Surface topography significantly influences the application processes, adhesion and strength of refinement coatings, as well as the aero- and hydrodynamic properties. It has a significant impact on subjective impressions of appearance, as well as on the preferences of purchasers of certain products. 

Surface topography is therefore one of the most important decision-making aspects in the manufacture of material components. Over the past several years, there have been remarkable advances in methods for measuring and processing data characterizing surface stereometry [40,41]. This makes it possible to greatly expand the scope of required analysis and the domain of choices that must occur in the design and manufacturing process.

The selection of parameters for evaluating the stereometric characteristics of technical surfaces, forming a complementary set that ensures high classification efficiency and ease of interpretation of assessments for specific surface applications, is a difficult task, requiring comprehensive analyses (Figure 8).

The surface of technical components is not just a geometric object, and although it is subject to digitization in the measurement process, it is a “material” object, and therefore its properties and purpose are important considerations for many analyses. The designer usually also determines the physical and chemical characteristics of the surface layer.

The basis for the selection of parameters to be used in evaluating a specific surface should be the purpose of the component and its operating conditions, while knowledge of the processes used to shape the surface should not be overlooked [40].

The shaping of the surfaces of many precision components is most often conducted by abrasive machining processes. Surfaces shaped in this way have randomized fractal features, with sometimes an “extraneous” principal component. At the same time, the fractal dimension of surface hills is higher than the fractal dimension of dales.

The information content of individual geometric structure parameters varies considerably [4,41]. The shortcomings of the structural design processes to date include the use of parameters with low information content and poor differentiation of geometric structure information in the definition of surface features most often. At the same time, there is an unfavorable great differentiation of the level of defining surface features in the design process and the level and possibility of surface evaluation, resulting from the knowledge and technique of data processing and the capabilities of the measuring devices [42].

It is worth noting that many parameters only gain significance after the information they contain is integrated with information from other parameters [43]. Most usable surfaces are designed to cooperate with other surfaces, so the distribution, sizes and statistical characteristics of potential contact fields are of great importance [44].

For the evaluation of surface quality, classification and interpretation of the result of geometric structure shaping, the ease of interpretation of parameter values and their reference to features and effects of the manufacturing process is of great importance [45].

It should not be expected that it is reasonable to indicate a single universal set of parameters recommended for evaluating the stereometric characteristics of the surfaces of designed components with different purposes, utility functions and operational applications.

Depending on the operating conditions defined during the design process and taking into account the features of the grinding process, such a set of parameters should be selected that will

maximize informational usefulness [45,46],meet the condition of complementarity [47],contain information about the dispersion and variability of geometric parameters [46,47],meet the condition of easy-to-interpret relations between the values of the parameters and the specified features of the surface [45].

In the diagnosis of grinding processes and in the evaluation of their results, it should be taken into account that many of the known parameters of surface topography do not show sufficient correlation with operational characteristics such as statistical features of potential contact zones of mating surfaces, their distribution, size and number, and the development of the perimeter of motifs resulting from surface irregularities.

Paper [47] presents the basis for evaluating the classification ability of surface geometric structure parameters and the basis for selecting known parameters. New parameters with higher classification ability were also developed.

In the advanced evaluation of the geometric structure of the surface, it is recommended to analyze the values and features of the gradient distribution, the distribution of the ordinates of surface vertices and outlines, the distribution of the distance of vertices in a certain direction, the size, position and distance of the fields of probable contact with the mating surface, and the geometric features of the motifs.

## 5. Classification Ability of Parameters Used to Evaluate the Geometric Structure of the Surface

Surfaces of machined workpieces may differ, to varying degrees, in the shape and distribution of irregularities, global and local features relating to height, shape, gradient, distribution and shapes of specific motifs, amplitude of individual harmonic components, degree of surface development, arrangement of machining marks, density of roughness vertices, sharpness and height of elevations, micro- and nano-structure of the surface.

The authors conducted a study of the contact area of mating surfaces associated with parallelism and perpendicularity of machining traces formed in the process of fine grinding. It was shown that the number of contact fields for perpendicular machining trace associations is small for small surface approximations and then increases significantly for larger approximations. The number of contact fields for parallel machining trace associations is quite large for small surface approximations and then increases to a lesser extent for larger approximations. An advanced basis for analyses of rough surface contact is presented in work [48].

The maximum number of contact areas for perpendicular machining trace associations ranges from 140 to 285 mm^−2^, and for parallel machining trace associations it does from 115 to 220 mm^−2^. It was also shown that the ratio of the area of the surface to the volume of the material under that surface, which describes the development of the geometric structure of the surface, is correlated with the number of contact fields. The greater the average absolute value of the surface gradient, the greater the number of contact fields. 

Recommendations for maximizing the contact stiffness of joints are derived from the above relationships. Smoothing surfaces with small form deviations, causing a reduction in the amplitude of high-frequency components (large gradient) reduces the number of contact fields, but significantly increases the contact fields. Then, perpendicular association of machining traces can be beneficial. 

Among the many parameters for evaluating geometric features of surfaces, there are many that do not differentiate individual surfaces in the analyzed set or do not distinguish a significant group. There are also such parameters whose values for each or most surfaces are significantly different, which means that they have a high ability to differentiate surfaces with respect to only certain topographic features. Therefore, it can be concluded that the quality of differentiating features of individual surfaces in the evaluated set increases with the increased number of parameters with high classification ability in it. 

An extensive set of known standards defined in several international guidelines was analyzed, with local variations based on national or sectoral standards. Standards to be cited include: ISO 4287, ISO 12085, ISO 13565-2 and ISO 13565-3, ASME B46.1, VDA 2006 VDA 2006, VDA 2007. A numerous set of new proprietary parameters for evaluating the geometric structure of surfaces was included in the analysis. A total of 83 parameters were analyzed. Taylor Hobson’s Talysurf CCI 6000 (Taylor Hobson, Leciester, UK) measurement system was used for the study.

The system has a resolution of 10 picometers over a measurement range of 100 µm. The repeatability for average values is 0.003 nanometers (3 picometers). An advanced type of measurement interferometer uses a patented algorithm to detect the position of the coherence peak of the interference pattern produced by a light source with a selectable bandwidth. This method provides high resolution interference images for all surfaces with reflectance from 0.3% to 100%. 

The measurement area is up to 300 mm × 300 mm with a resolution of 0.4 µm in the x and y directions. Measurement areas up to 7.2 mm × 7.2 mm can be measured without merging operations. Larger areas are combined into uniform sets. The maximum measurement space holds more than 500 billion points. Automatic procedures for analyzing the results were carried out using TalyMap^®^ 7 Platinum (Taylor Hobson, Leciester, UK) software and proprietary packages in the Matlab environment for analyzing surface topography.

A method was developed for determining the classification ability of specific parameters used to evaluate stereometric features of surfaces for a specific set of compared surfaces (normalized for a given set). It was shown that the more uniform the distribution of differences of normalized (in the interval 〈0÷1〉) values of a given parameter in an ascendingly ordered set for all surfaces, the greater the ability to differentiate the geometric structure of the surface. 

If a significant number of differences of normalized values for a particular parameter have very low values for the studied set of surfaces, the parameter has low classification ability.

An effective indicator of classification ability is the geometric mean of the differences of successive values of a given parameter, in a sorted set of values, normalized to the interval 〈0÷1〉 for a specific set of evaluated surfaces.

Recommendations [46] on the decision of selecting parameters for evaluating the geometric structure of a surface based on analyses for a set of parameters (83) and a set of surfaces (22) with similar values of the height of inequality are as follows:

In the first stage, a set of surfaces that are planned to be differentiated using different parameters is created.In the next stage, the values of various parameters are determined for all surfaces in the test set.Normalization of the parameters to the interval 〈0÷1〉 is carried out, advantageously using fuzzy inference methods.Visualization of normalized parameter values on a radar chart is performed.A sorting of sets of values of each parameter separately {*St*1, *St*2, ..., *Stn*}, ... {*P*1*i*, *P*2*i*, ..., *Pni*} is performed.The difference between successive values for each set of parameters $\Delta P_{j_{i}}=P_{j_{i+1}}-P_{j_{i}}$ is determined.A small value $\varepsilon \;\ll \;\Delta {\bar{P}}_{j_{i}}$ is determined, e.g., $\varepsilon =\left(0.01÷0.1\right)\;\Delta {\bar{P}}_{j_{i}}$.

The geometric mean in a set of n values of ∆*P_j_*_,*i*_+*ε* is determined according to the following formula:(1)$$Sg={\left(\underset{n}{\prod }\left(\Delta P_{ij}+\varepsilon \right)\right)}^{\frac{1}{n}},\;for\;each\;j=1,\ldots ,k.$$

The value of Sg<1 is a determinant of the classification ability of parameter *j*. Classification ability is higher the higher the value of Sg. High values of the geometric mean *Sg* of the ∆*P_ij_* increment, the values of a given parameter, normalized to the interval 〈0÷1〉 in an ascending order list of these values showed the parameters S5p, Sp, S10z, and from the group of new parameters, the parameters concerning the dispersion of motif field values above a certain level and the distance between flat elevations, in addition to parameters Sq, Sr2, Skk, Sdc, Sci, Sa, Vmp, Vm, Str, Vmc, Sku, Smc.

High classification ability in the tested set of surfaces showed new parameters, including

σ(sqrt(Pw)/mean(sqrt(Pw))—the ratio of the standard deviation of the root of the elevation area to the root of the mean elevation area;Lx, Ly—average distance of elevations in x and y directions;LwmaxX, LwmaxY—the largest elevation distances;Lxpl, Lypl, and LwplmaxX, LwplmaxY—average distances of flat elevations and maximum distances between the centers of flat elevations in the x and y directions (Figure 9).

The high differentiation potential of the parameters mentioned above can be explained by the high influence on the evaluation result of such features as the position, shape and height of the motifs formed by the highest surface vertices. 

This means that the study of parameters describing potential contact zones of mating surfaces provides a lot of valuable information for the evaluation of the geometric structure of surfaces. Defining the requirements for the geometric structure of a surface and selecting the parameters to be determined in the inspection process requires the following methodology:First, a set of different applications and operating conditions of the surface is defined.Then, the set of parameters used to describe the stereometric features of the surface is defined.A correlation check is performed between the parameters forming the set, taking into account that many typical parameters often adopted in the design process are correlated.In the next step, patterns of stereometric features typical of certain applications, requirements and operating conditions are determined.In the final step, using an artificial Hamming neural network, a qualification of the surface under evaluation is performed to one of the patterns, which means assigning a specific classification discriminant.

## 6. Assessment of Accuracy in Grinding Processes

### 6.1. Probabilistic Basis for Describing the Inaccuracy of Forming Workpieces in Grinding Processes

In paper [48], on the basis of discrepancy analyses of assessments of the impact of the tolerances adopted by the designer on the final result of the object’s accuracy, it was shown that there are problems in tolerating material boundaries in many designed components. This is especially the case for such shapes that are not composed of elementary fragments with simple geometry (e.g., modified serrations, pump components, turbines, and even streamlined bodies).

Humienny [49] rightly stated that international standards in the field of product geometry specifications lag behind the development of knowledge and the scientific basis of design and new developments in metrology. On the models developed, he showed that the surface interpretation of maximum and minimum material requirements is justified, as two accuracy requirements (dimensional tolerance and geometric tolerance) are transformed into one combined requirement. 

Design, manufacturing, process diagnostics and quality control, as well as data processing procedures and algorithms are inseparable and largely overlapping phases of manufacturing processes with many interdependencies, and the effects of decisions in these phases of production are interdependent.

When considering the inaccuracy of the dimensions and shape of workpieces, it is usually not possible to stop at determining measures of the location and variability of the statistical set of results corresponding to the investigated realizations of manufacturing tasks. This is because what is important is not so much the distribution of the results of a completed series of products, but rather the distribution of the results of all possible realizations.

The basic problem, before determining a way to assess accuracy in grinding processes, is to formulate a correct hypothesis about the form of the distribution of the values of the considered feature and to correctly verify this hypothesis. The analysis carried out takes into account that these distributions are the result of the composition of the interactions of many causes of deviations.

There can be many hypotheses about the distributions of various characteristics of inaccuracy, as many as there can be different distributions. The problem of choosing the right model for the distribution of the trait under consideration in the general population cannot be solved solely on the basis of searching for a model that is consistent with a given data set. It is necessary to analyze the mechanism of formation of a particular deviation in dimension or form. Then, the probability of the model not conforming to the results in subsequent realizations is reduced, especially for changed conditions.

If the deviation of dimension or form is the result of a grinding process that can be considered a stationary stochastic process, then the distribution of the inaccuracy characteristic under consideration in the set resulting from the implementation over a long period of time is subject to the same laws as the instantaneous distributions. Often, however, deviations in dimension and form are the result of processes that cannot be considered stationary. The aggregate distribution of these deviations is, as a result of the non-stationarity of the process, different from the instantaneous distribution and depends on the length of the period and on the time when data collection began.

In many analyses of the selection of accuracy levels in grinding processes, considerations are most often limited to the consideration of factors dependent on and independent of the parameters of the machining processes, and to the determination of the relationship between these factors and their effects.

On the basis of the analysis of a number of grinding processes, a methodology for proceeding was formulated, which includes the following sequence of research procedures:Assessing the causes of deviations and the locations where they occur in the technological system.Conducting an analysis of the statistical characteristics of the causes of deviations and evaluating whether the deviation:
is a random variable of step or continuous type,is a realization of a stationary or non-stationary random process,contains a non-random component, fixed or time-varying,is characterized by a large coefficient of variation, which is the ratio of a specific measure of dispersion of the value of the variable to a measure of position,is a random variable, independent of other causes of deviation.
Conducting an analysis of the influence that the considered cause has on the formation of certain inaccuracy characteristics, in particular:
whether the influence of the considered cause is significant,what is the nature of the transmission of random and deterministic signals through the technological system,what are the probabilistic characteristics of the system’s response to the forcing that is the cause of the deviation.
Conducting an analysis of the mechanism of accumulation of the influence of a wide variety of causes on the resultant distribution of deviation.

In many cases, deterministic analysis of the causes of deviations in the position of the tool surface relative to the machined surface is sufficient to detect the main causes of inaccuracy of the machined workpiece. Such an analysis, however, does not make it possible to assess the influence of a variety of causes on the nature and measure of the dispersion of the value of the inaccuracy feature under study, which is particularly important when considering features characterized by a high coefficient of variation, such as, for example, deviations in form, pitch, and local or as a function of time variation of these features.

A deviation in dimension or form can result from the interaction of many factors with very small influences, none of which dominates the others. For this reason, the theoretical analysis of the interactions and accumulation of multiple outflows is particularly important. This is because it makes it possible to infer the existence of specific causal relationships.

### 6.2. Distribution of Dimensional and Form Deviations as a Result of the Composition of Multi-causal Interactions in Stationary Machining Processes

The specified deviation of dimension or form in the general case can be an arithmetic sum, geometric sum, product, a set of extreme values or a certain function of the effects caused by various causes of deviation. Considering the distribution of a certain inaccuracy feature as a result of the interaction of multiple causes in a process considered stationary, the following model cases are distinguished:The resulting distribution is the distribution of a sum of random variables, with one or more factors having a decisive influence, and others having a small influence, or none of the many factors dominating the others.The distribution of the inaccuracy characteristic under study is the distribution of the geometric sum of random variables.The deviation of dimension or form, as a random variable, is dependent on the product of a large number of other variables, or the main cause of the deviation is the result of a multiplicative mechanism.The resulting distribution is the distribution of the scalar or vector product of random variables.The distribution of the inaccuracy characteristic under study is the distribution of the extreme values of a certain random quantity.

When the resulting distribution of a specific inaccuracy characteristic can be considered as a distribution of the sum of multiple random variables, then, as the number of these variables becomes large, the resulting distribution tends toward a normal distribution. However, certain conditions must be met regarding the independence of the random variables and their joint distribution. Usually, there are only a moderately large number of considered causes of inaccuracy. However, if none of these causes dominates over the others, and if the variables are not highly dependent, then the distribution of the sum of the effects is close to a normal distribution.

In many cases, it can be assumed that the formation of workpiece deviations in grinding processes is the result of the sum of elementary geometric effects. For example, the deviation in the length of a shaft or the deviation in the distance of the faces of successive steps in a shaft of different diameters can be provided here.

Its widespread familiarity, ease of use and the fact that the normal distribution forms the basis of many assumptions in statistical analysis make it the most widely used model for the distribution of random variables. Unfortunately, it is not always in accordance with the mechanism of cumulative random influences. Even if the normal distribution is a consequence of the laws governing the dispersion of the values of the inaccuracy characteristic under study, its adequacy for describing a set of data decreases as it moves away from the mean value. Thus, in particular, the hasty use of the normal distribution as a model for the dispersion of values significantly smaller or larger than the expected value, or the adoption of such a model without physical considerations, in an area with a small data set, can lead to significant errors.

When one or more causes dominate the others, the resulting distribution can be inferred to be a composite of the distributions of the dominant factors and the normal distribution. The normal distribution in this case approximates the sum of many small influences resulting from the other causes. 

As an example of such a situation, consider the process of grinding screw surfaces on a grinding machine with radial runout of the workpiece spindle. This runout, described, for example, by the Rayleigh distribution (Weibull distribution for k = 2), can dominate over the other causes of machining inaccuracy.

Otherwise, after grinding on grinders exhibiting increased backlash in the wheel-to-workpiece feed system, the sum distribution of the deviation of ground shafts can be expected to be a composite of a normal distribution and a uniform distribution. In the case of the dominance of one or more causes of deviation, the greater the dispersion of the effects caused by the dominant cause, the greater the error of approximating the distribution of deviation values by a normal distribution.

When machining rotating surfaces, circularity deviations are the sum effect of multiple radial elementary displacements of random magnitude and random direction. The total displacement is then equal to the geometric sum of the elementary displacements.

If the displacement vectors obey the law of normal distribution, or if it can be assumed that the sum components of the vectors in the x and y directions are sums of multiple partial components, and none of them dominates over the others, then the probability density of the modulus of the random component of the leading vector is described by the following formula:(2)$$f\left(\Delta R\right)=\frac{\Delta R\cdot e^{\left(\frac{-\Delta R^{2}\left(Dx^{2}+Dy^{2}\right)}{4Dx^{2}Dy^{2}}\right)}}{DxDy}J_{0}\left(\frac{\Delta R^{2}\left(Dx^{2}-Dy^{2}\right)}{4Dx^{2}Dy^{2}}\right),$$
where *J*_0_ is the Bessel function of zero order, ∆*R* is the random component of radial displacement, *Dx* and *Dy* are the variances of the components of displacement in *x* and *y* directions.

When the systematic deviation can be considered zero, the radial runout can be described by a Rayleigh distribution. This distribution can be used to describe the deviations of eccentricity, parallelism, as well as point distances, which are the geometric sum of multiple elementary deviations.

In analyses of grinding accuracy, it is often necessary to determine the permissible deviations arising under the multiplicative effects of the effects of various causes of inaccuracy. 

A common case of a multiplicative mechanism for shaping deviation values during grinding is the sparking process itself. Each sparking pass, assuming the linear characteristics of the technological system, reduces the deviation of dimension or shape in a random manner, in proportion to its value. The value of the deviation after the *n*-th sparking pass is equal to the product of that deviation before that pass (*n* − 1) and the random factor of the deviation reduction.

The resulting random variable, and in this particular case the resulting deviation in dimension or shape when the number of sparking passes is large, is the product of a significant number of other variables, and it is impossible to examine and describe each one.

However, it is possible to determine the resulting distribution, provided that the conditions accompanying the central limit theorem are satisfied. The dispersion of deviation values ∆*_n_* is then described by a log-normal distribution:(3)$$f\left(\Delta_{n}\right)=\left\{\begin{array}{cc} \frac{1}{\sqrt{2\pi }\Delta_{n}D\left(ln\left(\Delta_{n}\right)\right)}e^{\left(-\frac{{\left(ln\left(\Delta_{n}\right)-\mu ln\left(\Delta_{n}\right)\right)}^{2}}{2D{\left(ln\left(\Delta_{n}\right)\right)}^{2}}\right)} & ,x>0\\ 0 & ,x\leq 0 \end{array}\right..$$

The log-normal distribution does not have the additive property of regeneration, so the sum of random variables with log-normal distributions does not have the same form of distribution. In contrast, the product of random variables with log-normal distributions is characterized by a distribution of this type as well.

In assessing the inaccuracy of ground parts, it is often the case that the greatest deviation in dimension or form determines the quality. This is the case when evaluating tool edge roughness or the roughness of the edges of other components such as thin ceramic backing plates. In long screws, the essential feature of inaccuracy may be the largest value of the sum of the pitch deviation, in gears the largest deviation of the sum of any number of pitches, and in long shafts the largest roundness deviation.

In the cases mentioned above, the largest value of *x_m_* among n random variables *x*_1_, ..., *x_n_* is sought. If the variables *x_i_* meet the assumptions of independence and joint distribution, then the distribution of *x_i_* values has relatively little effect on the distribution of the variable *x_m_* = max(*x_i_*). The above assumptions are, for the form deviations of abrasive machined surfaces, satisfied more often than for surfaces machined by other methods.

In such cases, it is possible, for a significant number of forest variables, to determine a limiting distribution that will well describe the behavior of the *x_m_* variable, even if the distribution of the *x_i_* variables is not known. Three of the best studied special cases will be discussed below.

The distribution of the largest values of Type I is obtained for the assumption that the probability of indefinitely increasing values of the variables *x_i_* decreases exponentially. This means that the probability density is of the following form:(4)$$f\left(x\right)=\frac{d\left(g\left(x\right)\right)}{dx}e^{\left(-g\left(x\right)\right)},$$
where *g*(*x*) is an increasing function of the variable *x*. 

The probability density of the maximum values is then obtained:(5)$$f\left(x_{m}\right)=\alpha e^{-\alpha \left(x_{m}-u\right)-e^{-\alpha \left(x_{m}-u\right)}},$$
where *u* is the mod of the distribution of the variable *x_m_*, and α is a measure of dispersion. 

The expected value and variance are then expressed by approximate relationships: E (*x_m_*) = *u* + 0.577α^−1^; D^2^ (*x_m_*) = 1.645α^−2^; the coefficient of asymmetry is then equal to 1.1396.

Type II maximum value limit distribution occurs when the random variable is bounded from below by zero and unbounded from above. Then, the probability density functions of the *x*-values and *x_m_*-maximum values have the following form:(6)$$f\left(x\right)=\beta kx^{-\left(k+1\right)}\;for\;x\geq 0,$$
(7)$$f\left(x_{m}\right)=ku^{k}x_{m}^{-\left(k+1\right)}e^{-{\left(\frac{u}{x_{m}}\right)}^{k}}for\;x\geq 0.$$

The relationship between a Type II distribution and a Type I distribution is the same as that between a log-normal distribution and a normal distribution. If *x_m_* has a Type II distribution, then ln(*x_m_*) has a Type I distribution. The limit distribution of the largest values of Type III corresponds to the restriction of random variables from above and is known as a Weibull distribution.

### 6.3. Momentary Distribution vs. Summary Distribution of Dimensional and Form Deviations of Ground Components

In manufacturing processes, due to many reasons such as deformation of the technological system, dependent on the position of the machining zone, machining forces, varying temperature fields and tool wear, deviation distributions are not the realization of stationary processes [12,19,50,51,52,53,54]. 

Typically, there is a movement of the center of the grouping of deviation values (Figure 10). The application example presented here is of an automated grinding process for small ceramic shapes using a grinding wheel with a hyperboloidal active surface (Figure 11) and the device shown in Figure 12. In this example, a repeatedly extended grinding zone is achieved, as well as a decreasing rate of allowance removal as the workpieces move along the machining zone. 

The summary distribution of the flatness deviation from the 300 min period is an asymmetric distribution, and the location of the frequency maximum depends on the trend of the change in the average value. On the basis of the features of the dispersion of the deviation value in the summed set of elements, it is not possible to infer its causes. Only the determination of the “momentary” values of the deviation dispersion parameters allows a more complete diagnosis of the process.

In other realizations of grinding processes, there may also be a change in the dispersion measures. Then, the aggregate distribution is modified by two mechanisms: one, described above, increases the range of variation and flattens the form of the distribution, and the other, on the contrary, makes it slender.

The probability that at time t the deviation value *x* is within *x*, *x* + *dx* is equal to *f*(*x*,*t*)*dx*. If the probability that any taken deviation value, formed in the period *t*, *t* + *dt* is *ψ*(*t*)*dt*, then the summary distribution of the deviation is expressed by the following relation:(8)$$F\left(x\right)=\overset{t_{0}}{\underset{0}{\int }}\psi \left(t\right)dt\overset{\infty }{\underset{-\infty }{\int }}f\left(x,t\right)dx.$$

In conclusion of the analyses, concerning the modeling of distributions of dimensional or form deviations, it should be emphasized that tests of the conformity of the distribution with a specific model are not, of course, an assessment of conformity with the physical characteristics of the dispersion of a given size. The tests only answer the question of whether the experimental data contradict the adopted model. 

In defining requirements for the accuracy of designed components, using models of the distribution of specific inaccuracy characteristics, particular care should be taken in evaluating marginal zones of variation. There can be significant differences in these zones, which are of great importance for the quality of products, even when the model shows high agreement with data grouped near the central values.

## 7. Conclusions

Grinding processes are characterized by a significant dispersion of values of quantities characterizing the course and results of machining. The distribution of the studied features depends on the mechanism of cumulative effects of random disturbances. In most cases, there is a multiplicative mechanism or, with regard to geometric features, an additive vector mechanism. 

The high energy intensity of processes results in significant forces, the generation of unfavorable stress states in the surface layer, the occurrence of extensive fields of elevated temperatures, which causes deformation of the workpiece, tool and machining system. The properties of the surface layer and the accuracy of machining are affected not only by the energy intensity of the processes, but also by the local concentration of energy, the shape and location of the machining zone. 

Comprehensive diagnostics of grinding processes include many features, the analysis of which requires consideration of the probabilistic basis of inaccuracy in the shaping of workpieces, random variation, fluctuations and complex mechanisms of phenomena in abrasive tool wear processes. 

A method for determining the classification ability of specific parameters used to evaluate stereometric features of ground surfaces is presented. It is shown that the more uniform the distribution of normalized differences (in the interval 0, 1) of the values of a given parameter in an ascending ordered set for all compared surfaces, the greater the ability to differentiate the geometric structure of the surface.

An effective indicator of classification ability is the geometric mean of the differences of successive values of a given parameter, in a sorted set of values, normalized to the interval <0.1> for a specific set of evaluated surfaces.

In the analysis of grinding accuracy, in many cases it is necessary to determine the permissible deviations, formed under the conditions of multiplicative interaction of the effects of various causes of inaccuracy. 

The multiplicative mechanism of forming deviation values during grinding is typical of the sparking process. Each sparking pass causes a reduction in the deviation of dimension or form in a random manner, but depending on its previous value. The dispersion of deviation values is then described by a log-normal distribution. In many other cases, quality is determined by the largest deviation of dimension or form, in which case distributions close to the limit distributions of the largest values should be expected.

The dispersion of deviation values is then described by a log-normal distribution. In many other cases, quality is determined by the largest deviation in dimension or shape, in which case distributions close to the limit distributions of the largest values should be expected.

In manufacturing processes, due to many reasons such as deformation of the technological system, dependent on the position of the machining zone, machining forces, varying temperature fields and tool wear, deviation distributions are not the realization of stationary processes. 

Therefore, on the basis of the characteristics of the dispersion of deviation values in the sum set of elements, it is not possible to infer their causes. Only the determination of the “momentary” values of the deviation dispersion parameters allows a more complete diagnosis of the process.

## Figures and Tables

**Figure 1 materials-16-02195-f001:**
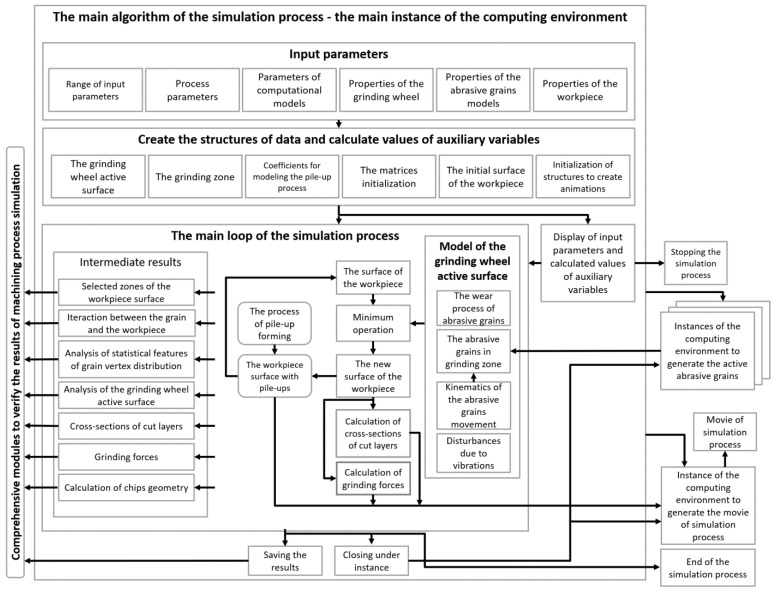
Schematic diagram of the structure of a comprehensive system for simulating the grinding process.

**Figure 2 materials-16-02195-f002:**
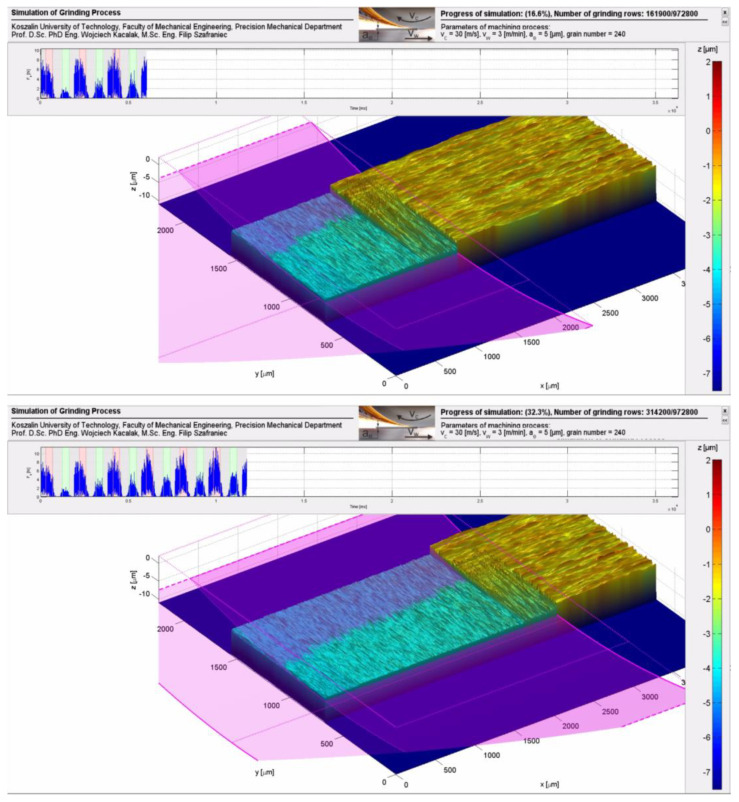
Visualization of selected surface topography states and force variation during the grinding process.

**Figure 3 materials-16-02195-f003:**
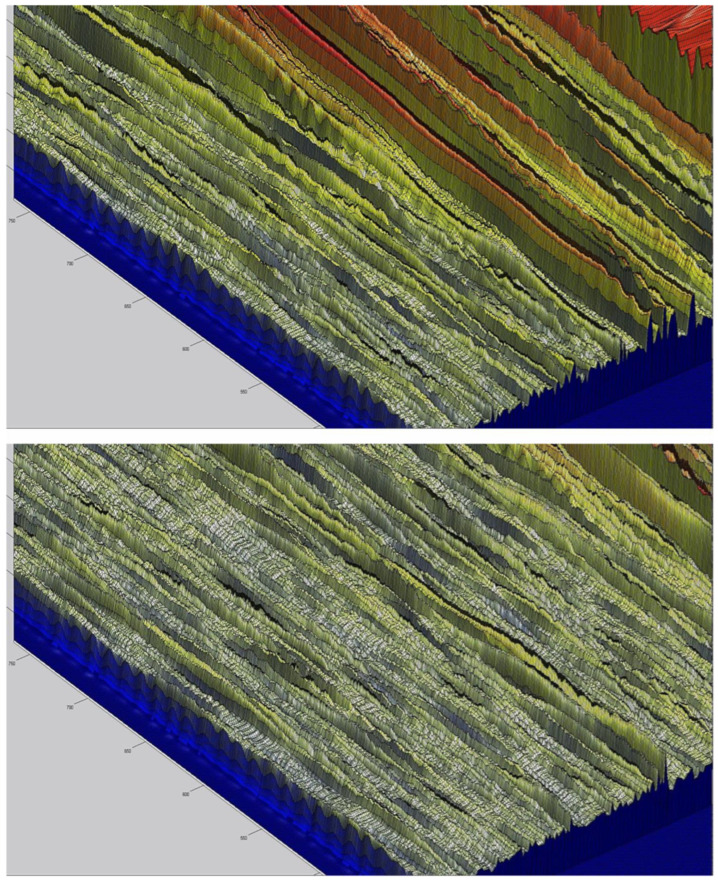
Images of the topography of the machined surface during grinding of planes with a traverse feed of 0.3 mm; after two—traverse feeds-top image, after three—traverse feeds-bottom image.

**Figure 4 materials-16-02195-f004:**
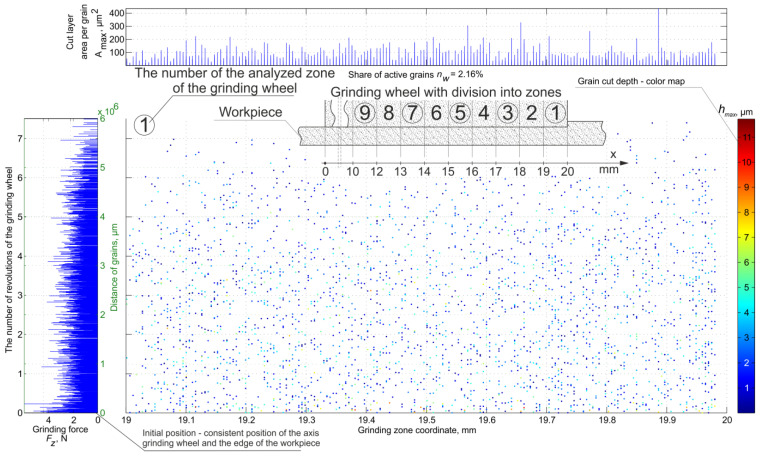
Map of grain activity in the first zone during plane grinding with traverse feed.

**Figure 5 materials-16-02195-f005:**
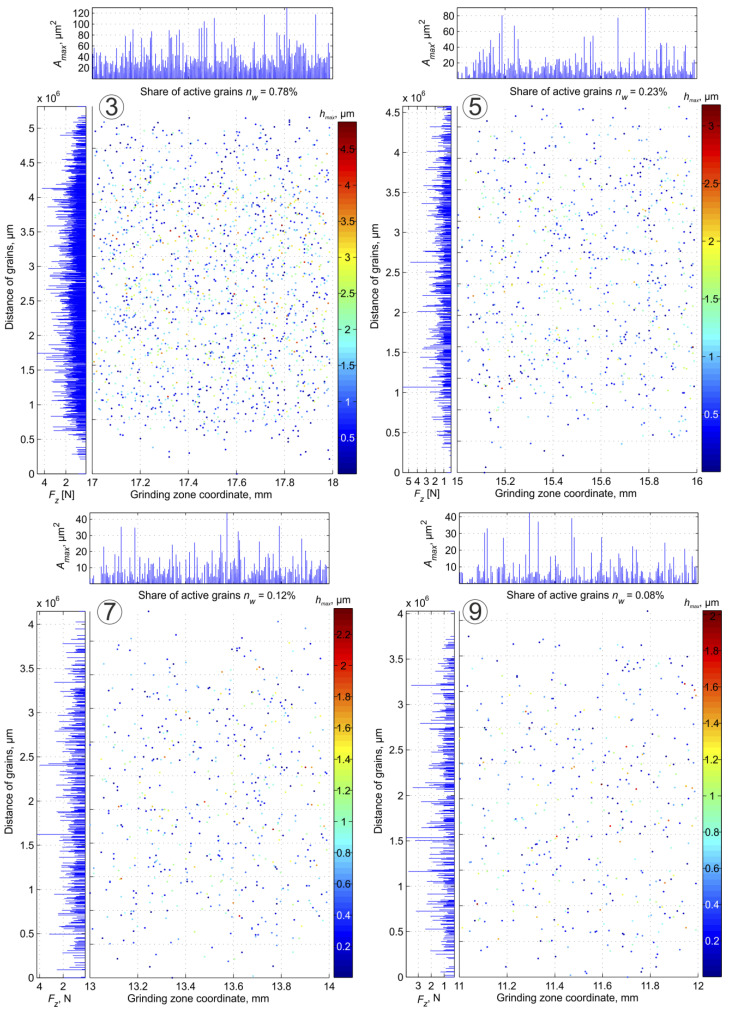
Maps of grain activity in Zones 3, 5, 7, 9 (according to Figure 4)—the width of the zone is equal to the value of the traverse feed rate.

**Figure 6 materials-16-02195-f006:**
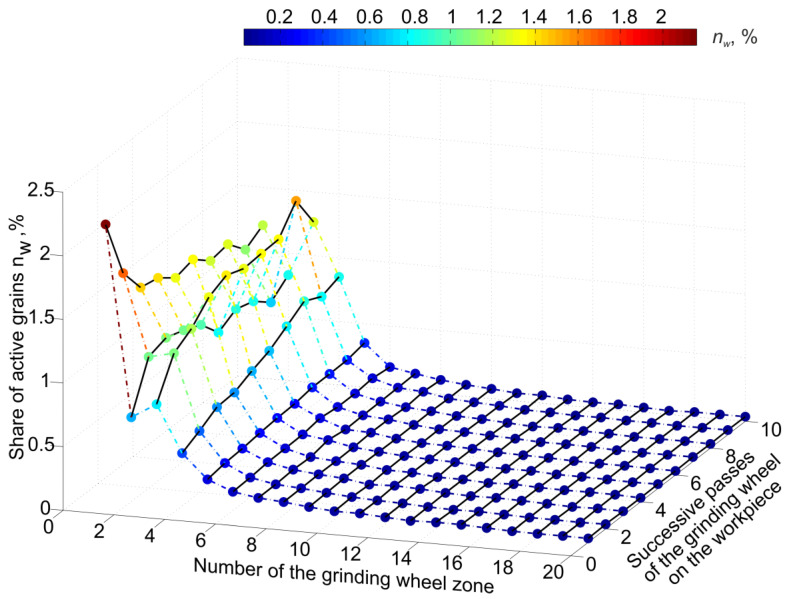
Share of active grains in the total number of grains on the grinding wheel surface depending on the zone numbers.

**Figure 7 materials-16-02195-f007:**
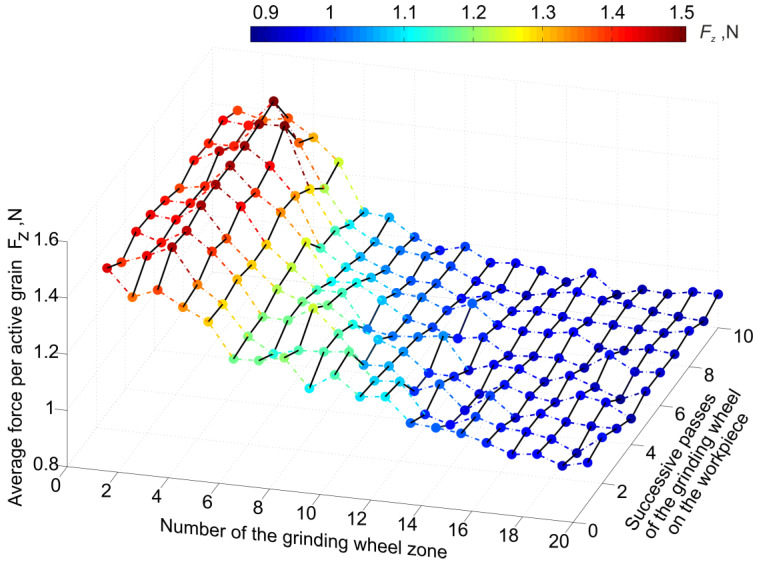
Average force per active grain depending on the zone number on the grinding wheel surface.

**Figure 8 materials-16-02195-f008:**
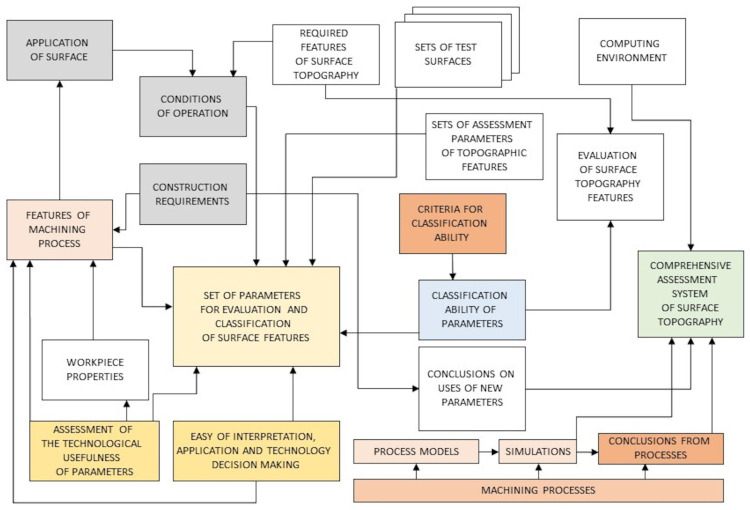
A scheme of the system of comprehensive evaluation of geometric characteristics of surfaces in terms of decisions in the process of design and manufacture of technical components.

**Figure 9 materials-16-02195-f009:**
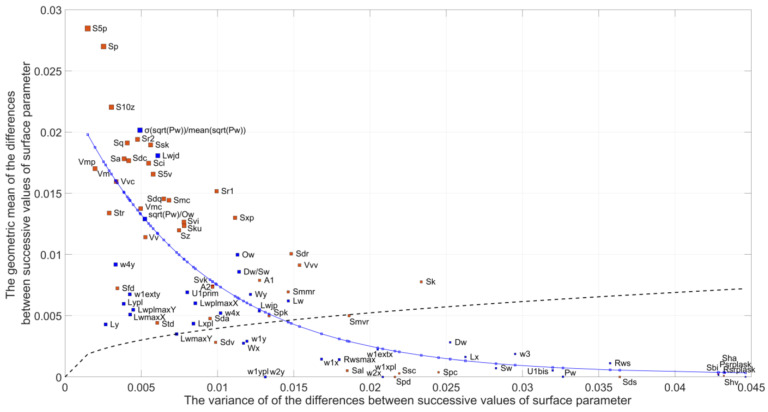
Dependence of the geometric mean of the differences of successive values of the analyzed parameters (determined for each plot from the adopted set) on the variance of these differences.

**Figure 10 materials-16-02195-f010:**
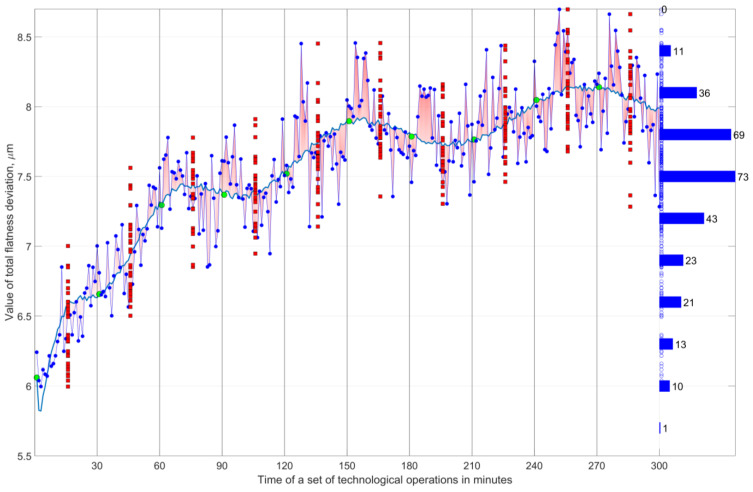
Results of the study of the form of momentary and total flatness deviation distribution of ground small ceramic parts in the automated grinding process using the built AR7 machining system (depicted on Figure 11 and Figure 12).

**Figure 11 materials-16-02195-f011:**
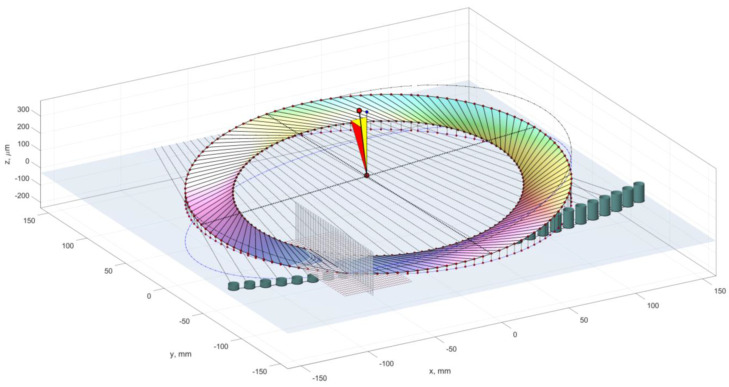
Grinding method for small ceramic parts.

**Figure 12 materials-16-02195-f012:**
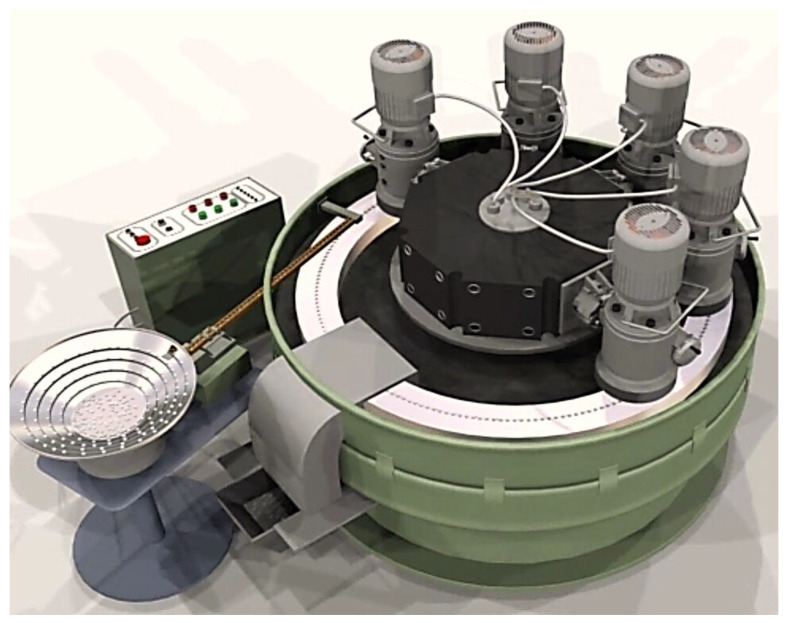
Automatic grinding machine for processing small ceramic parts (as one of a series of devices designed and made in the Department of Technical and Information Systems of Koszalin University of Technology).

## Data Availability

Not applicable.

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
