# Peer review of "Metrological Aspects of Assessing Surface Topography and Machining Accuracy in Diagnostics of Grinding Processes"

_materials, 2023, doi:10.3390/ma16062195_

Round 1

Reviewer 1 Report

This paper introduces the probability of grinding process diagnosis, and considers the metrological aspects of machining surface topography evaluation and the selection of machining accuracy evaluation. The research content is very meaningful. I think this paper can be published in this journal. However, minor revision are required before acceptance and publication.

1. The writing format of the paper should preferably be composed of four parts. For example, the first part is introduction. The second part is methods and materials. The third part is discussion and analysis. The fourth part is the conclusion. This paper is divided into 9 parts, and the middle parts can be summarized in the discussion and analysis of the third part.

2. In Parts 5 and 6, the same serial number should no longer be used, such as 1. 2. 3. etc. It should be replaced with other serial numbers.

Author Response

Dear Reviewer, thank you for considering our manuscript for publication in Materials. We are grateful for the valuable suggestions provided. In the revised manuscript, all changes are highlighted in red.

Below, we submit responses to the comments.

Point 1: The writing format of the paper should preferably be composed of four parts. For example, the first part is introduction. The second part is methods and materials. The third part is discussion and analysis. The fourth part is the conclusion. This paper is divided into 9 parts, and the middle parts can be summarized in the discussion and analysis of the third part.

Response 1: The “Materials and method” chapter has been added. The number of chapters have been reduced from 9 to 7.

Point 2: In Parts 5 and 6, the same serial number should no longer be used, such as 1. 2. 3. etc. It should be replaced with other serial numbers.

Response 2: It has been changed.

The authors thank the Reviewer for their time spent analysing the manuscript and valuable comments.

Reviewer 2 Report

Though the manuscript took a very interesting perspective in diagnosing grinding processes, the content does not support its claim.

For example, the kinematic simulation the manuscript presented is only a demonstration of the simulation itself. No parameters or modeling process were introduced and explained. The result does not involve any meaningful conclusion which can support the "importance of simulation".

Overall, the manuscript feels more like a review, but it lacks in depth analysis of the current studies. 

In my opinion, I don't think this manuscript is suitable for publishing in its current stage. 

Author Response

Dear Reviewer, thank you for review our manuscript for publication in Materials. We are grateful for the valuable suggestions provided. In the revised manuscript, all changes are highlighted in red.

Below, we submit responses to the comments to the review.

Though the manuscript took a very interesting perspective in diagnosing grinding processes, the content does not support its claim.

For example, the kinematic simulation the manuscript presented is only a demonstration of the simulation itself. No parameters or modeling process were introduced and explained. The result does not involve any meaningful conclusion which can support the "importance of simulation".

Response: The manuscript has been the results of simulation tests chapter added. The input parameters and ranges of results for analysis included in research has been described.

The authors thank the Reviewer for their time spent analysing the manuscript and valuable comments.

Reviewer 3 Report

The authors presented issues of probabilistic diagnostics of grinding and evaluation of topography of machined surfaces with machining accuracy assessment. The processes of creating the geometric structure on the grinded surface are described and introduces a method for establishing the classification ability of specific parameters for evaluate stereometric properties of grinded surfaces. Also was emphasized that based on the characteristics of the deviation value dispersion is not possible to conclude its causes but the determination of the "instantaneous" values of the deviation dispersion parameters will be allowed a more complete diagnosis of the process.

The paper is remarkably interesting, and I have no fundamental doubt in the presented research.

I found some errors in this manuscript, and it must be improved.

Strength

The work is remarkably interesting especially in terms of the extensive simulation of the grinding process.

Noticed errors

1.       In the introduction chapter, it is necessary to take a slightly broader look at the use of abrasive grains in modern processing technologies, also other than pure grinding, such as Abrasive Water Jet, for example in the topic of experimental research into marble cutting by abrasive water jet and/or the quantitative evaluation of the cutting surface quality levels in abrasive water jet cutting by measurement of the representative striation mark displacement.

2.       Lines 65 and 80. Bulk citation of more than 2-3 sources in the analysis of state of art is not a good practice. Authors should expand the analysis of the state of the issue and refer to each cited item separately (or to 2-3 items group) in a few sentences.

3.       In chapter 1. Introduction lacked a summary of the state of the issue, a clear presentation of the research gap and succinctly stated purpose of this paper

4.  I have the irresistible impression that the authors somewhat erroneously but consistently use the term "transverse feed” and should use "traverse feed". 

Small errors

These errors do not diminish the value of this interesting work, but need to be improved

1.       Variables in equations written in italics should also be in italics in the descriptions in the text of the paper. This should be carefully checked throughout the paper.

2.       Line 56. Is: J/mm3; should be: J/mm3

3.       Lines 440, 428, 755. Interval notation should be standardized.

4.       Line 435. Is (0,01 ÷ 0,1); should be: (0.01 ÷ 0.1);

Author Response

Dear Reviewer, thank you for considering our manuscript for publication in Materials. We are grateful for the valuable suggestions provided. In the revised manuscript, all changes are highlighted in red.

Below, we submit responses to the comments to the review.

The authors presented issues of probabilistic diagnostics of grinding and evaluation of topography of machined surfaces with machining accuracy assessment. The processes of creating the geometric structure on the grinded surface are described and introduces a method for establishing the classification ability of specific parameters for evaluate stereometric properties of grinded surfaces. Also was emphasized that based on the characteristics of the deviation value dispersion is not possible to conclude its causes but the determination of the "instantaneous" values of the deviation dispersion parameters will be allowed a more complete diagnosis of the process.

The paper is remarkably interesting, and I have no fundamental doubt in the presented research. […] The work is remarkably interesting especially in terms of the extensive simulation of the grinding process.

Thank you for your positive opinion about our article.

Point 1: In the introduction chapter, it is necessary to take a slightly broader look at the use of abrasive grains in modern processing technologies, also other than pure grinding, such as Abrasive Water Jet, for example in the topic of experimental research into marble cutting by abrasive water jet and/or the quantitative evaluation of the cutting surface quality levels in abrasive water jet cutting by measurement of the representative striation mark displacement.

Response 1: The 25 position of bibliography has been added.

Point 2: Lines 65 and 80. Bulk citation of more than 2-3 sources in the analysis of state of art is not a good practice. Authors should expand the analysis of the state of the issue and refer to each cited item separately (or to 2-3 items group) in a few sentences.

Response 2: The bulk citation has been separated for a few sentences.

Point 3: In chapter 1. Introduction lacked a summary of the state of the issue, a clear presentation of the research gap and succinctly stated purpose of this paper.

Response 3: From the line 133 to the end of the introduction chapter, has been presented a bulleted list of problems that should be addressed in order to comprehensively diagnose of grinding processes.

Point 4: I have the irresistible impression that the authors somewhat erroneously but consistently use the term "transverse feed” and should use "traverse feed".

Response 4: It has been corrected.

Point 5: These errors do not diminish the value of this interesting work, but need to be improved

  1. Variables in equations written in italics should also be in italics in the descriptions in the text of the paper. This should be carefully checked throughout the paper.
  2. Line 56. Is: J/mm3; should be: J/mm3
  3. Lines 440, 428, 755. Interval notation should be standardized.
  4. Line 435. Is (0,01 ÷ 0,1); should be: (0.01 ÷ 0.1);

Response 5: This small errors have been corrected.

The authors thank the Reviewer for their time spent analysing the manuscript and valuable comments.

Reviewer 4 Report

Dear Authors,

it would be important to have a separate chapter called "Methods and Materials" after the Introduction. The number of chapters is basically very large. I recommend that there be a single "Results" chapter and that the authors present the results in subchapters within it. The figures are beautiful and easy to understand. The literature used is professionally considered, although I think the number of self-references is a bit high (the publisher decides to what extent this is allowed, I don't want to get too involved). I admit I missed the Discussion chapter, but it is not mandatory in principle. I was happy to read this article, congratulations.

Author Response

Dear Reviewer, thank you for considering our manuscript for publication in Materials. We are grateful for the valuable suggestions provided. In the revised manuscript, all changes are highlighted in red.

Below, we submit responses to the comments to the review.

It would be important to have a separate chapter called "Methods and Materials" after the Introduction. The number of chapters is basically very large. I recommend that there be a single "Results" chapter and that the authors present the results in subchapters within it. The figures are beautiful and easy to understand. The literature used is professionally considered, although I think the number of self-references is a bit high (the publisher decides to what extent this is allowed, I don't want to get too involved). I admit I missed the Discussion chapter, but it is not mandatory in principle. I was happy to read this article, congratulations.

Response: The “Materials and method” chapter has been added. The number of chapters have been reduced from 9 to 7. Thank you for your positive opinion about our article.

The authors thank the Reviewer for their time spent analysing the manuscript and valuable comments.

Round 2

Reviewer 2 Report

After adding the methodology section, the manuscript feels much better than its original version.